# Evolutionary dynamics of dengue virus in India

**Suraj Jagtap**[1], **Chitra Pattabiraman**[2], **Arun Sankaradoss**[3], **Sudhir Krishna**[3,4], **Rahul Roy** [1,5]*

**1** Department of Chemical Engineering, Indian Institute of Science, Bengaluru, Karnataka, India, **2** Infectious Disease Research Foundation, Bengaluru, Karnataka, India, **3** National Centre for Biological Sciences, Tata Institute of Fundamental Research, Bengaluru, Karnataka, India, **4** School of Interdisciplinary Life Sciences, Indian Institute of Technology Goa, Ponda, India, **5** Center for BioSystems Science and Engineering, Indian Institute of Science, Bengaluru, Karnataka, India

* rahulroy@iisc.ac.in

**Data Availability Statement:** All relevant data are within the manuscript and its Supporting Information files.

**Funding:** R.R. acknowledges the support of The Wellcome Trust-DBT India Alliance (Intermediate

## Abstract

More than a hundred thousand dengue cases are diagnosed in India annually, and about half of the country's population carries dengue virus-specific antibodies. Dengue propagates and adapts to the selection pressures imposed by a multitude of factors that can lead to the emergence of new variants. Yet, there has been no systematic analysis of the evolution of the dengue virus in the country. Here, we present a comprehensive analysis of all DENV gene sequences collected between 1956 and 2018 from India. We examine the spatio-temporal dynamics of India-specific genotypes, their evolutionary relationship with global and local dengue virus strains, interserotype dynamics and their divergence from the vaccine strains. Our analysis highlights the co-circulation of all DENV serotypes in India with cyclical outbreaks every 3–4 years. Since 2000, genotype III of DENV-1, cosmopolitan genotype of DENV-2, genotype III of DENV-3 and genotype I of DENV-4 have been dominating across the country. Substitution rates are comparable across the serotypes, suggesting a lack of serotype-specific evolutionary divergence. Yet, the envelope (E) protein displays strong signatures of evolution under immune selection. Apart from drifting away from its ancestors and other contemporary serotypes in general, we find evidence for recurring interserotype drift towards each other, suggesting selection via cross-reactive antibody-dependent enhancement. We identify the emergence of the highly divergent DENV-4-Id lineage in South India, which has acquired half of all E gene mutations in the antigenic sites. Moreover, the DENV-4-Id is drifting towards DENV-1 and DENV-3 clades, suggesting the role of cross-reactive antibodies in its evolution. Due to the regional restriction of the Indian genotypes and immunity-driven virus evolution in the country, ~50% of all E gene differences with the current vaccines are focused on the antigenic sites. Our study shows how the dengue virus evolution in India is being shaped in complex ways.

Fellowship: IA/I/13/1/500901). R.R and S.K also acknowledge funding from philanthropist Mr. Narayana Murthy, Co-founder of Infosys (17X6777). C.P. acknowledges the support of The Wellcome Trust-DBT India Alliance (Intermediate Fellowship: IA/E/15/1/502336). S.J. acknowledges the support of the Ministry of Human Resource Development Fellowship. The funders had no role in study design, data collection and analysis, decision to publish, or preparation of the manuscript.

**Competing interests:** The authors have declared that no competing interests exist.

## Author summary

Dengue is a mosquito-borne disease with four closely related serotypes of the virus (DENV-1-4). Further, cross-reacting dengue antibodies from a previous infecting dengue serotype can protect or enhance infection from other serotypes. This can force the emergence of new dengue variants that find ways to escape the immune action or take advantage of it. In endemic countries like India, high rates of previous dengue infection can drive the evolution of dengue serotypes in complex ways. We compare all published dengue virus sequences to understand how new variants of dengue are emerging in India. Dengue cases and corresponding viruses display triennial surges. Further, the dengue envelope protein for each serotype shows recurring divergence and reversal towards its ancestral strain over a three-year window. Such fluctuations are also correlated among the dengue serotypes in India and could arise from the changing levels of cross-reactive antibodies. This, combined with the regional exchange of the virus among Asia-Pacific countries, has led to the emergence of India-specific DENV lineages, including a new DENV-4 (Id) variant. This has also contributed to significant variations in the epitope regions of the current dengue viruses in India compared to the vaccines with implications for their efficacy.

## Introduction

Dengue infections have increased dramatically in the last two decades and are expected to rise further as they spread to newer regions fuelled by urbanization and travel [1]. About half of the world population is at risk of dengue infections [1,2]. Estimates from 2010 claim 390 million annual dengue infections worldwide, of which only 96 million cases were reported clinically [3]. Around one-third of these infections were estimated to be from India [3], though most of them go unreported [4]. Dengue is endemic in almost all states in India [5,6]. All four antigenically distinct serotypes (DENV-1-DENV-4) of the virus that display significant immunological cross-reactivity due to 65–70% homology have been reported from various parts of the country [7–9]. Combined with a complex transmission cycle and high dengue seroprevalence [5], dengue evolution in the country has been shaped in complex and unexpected ways though this remains poorly understood.

Global dengue virus evolution is modulated by pathogen transmission bottlenecks and immunological pressures [10,11]. An increase in globalization and human mobility can lead to the global spread of the emerging dengue virus strains. On the other hand, the acute nature of the disease, limited travel range and restriction to tropical regions of the vector can constrain the virus spread. Like other vector-borne virus infections, the dengue virus switches its environment due to horizontal transmission, exerting a strong purifying selection pressure [10,12,13]. In the human host as well as at the larger population scale, pre-existing immunity can contribute to the emergence of immune escape variants. Heterotypic immunity can also shape the co-evolution of dengue serotypes contingent on the level of cross-reactive antibodies and the antigenic similarity between the infecting serotypes during primary and secondary infection [11,14]. Further, antibody-dependent enhancement (ADE) under sub-optimal levels of cross-reactive antibodies can confer a selective advantage to antigenically related serotypes [15–17].

Complex population immunity against the dengue virus can also modulate the levels of annual infections and caseload. Cyclic dengue outbreaks in the endemic regions occur every 2–4 years, often associated with serotype/genotype replacement, where serotype/genotype

dominance changes during the subsequent outbreak [18–21]. This has been attributed to a combination of long-term protection from homotypic dengue infection albeit short-term protection (up to 2 years) from the heterotypic secondary infection [22–24]. This allows the heterotypic serotype to manifest an outbreak depending on the prevalence of serotypes and population immunity in the region [18]. During serotype replacement, ADE can also play a role in increasing dengue cases. Increased dengue viremia levels observed during ADE [25–27] can contribute to increased transmission of the virus [28,29]. Therefore, ADE related effects can further make the cyclic pattern of outbreaks more prominent. However, whether this advantage also leads to the evolution of the virus, remains unknown. Therefore, knowledge of longitudinal prevalence, serotype distributions, and prior serotype of infection can help us in understanding the evolution of the dengue virus and predicting future outbreaks [30].

In spite of being a hotspot of dengue infections, the scarcity of dengue genomic data from India has limited our understanding of dengue virus evolution. Previous dengue studies in India have focused on regional outbreaks [7–9,31–33], which are dominated by single or closely related strains due to the limitations of a short collection period. The persistence of all serotypes in a high seroprevalence background has been shown to manifest in immunity-driven co-evolution of dengue virus strains at the city-scale [11]. In the absence of longitudinal analysis of dengue viral diversity, it remains unknown, whether such selection pressures are shaping dengue virus divergence at a country-wide scale in India.

Large divergence of prevalent dengue genotypes can have significant implications for vaccine design and development [34,35]. Multiple dengue vaccines targeting all four serotypes have been developed and are currently at different stages of clinical trials [36]. These vaccines are based on the old dengue isolates from outside South Asia. In the absence of efficacy studies in India, it remains unclear whether they will induce optimal levels of neutralizing antibodies against the dengue viruses circulating in India. Apart from not providing sufficient protection against dengue infection, some vaccine candidates can even lead to enhanced disease through ADE upon subsequent infection, as seen in the case of the CYD-TDV vaccine [37,38]. Yet, differences at the antigenic sites between vaccines and prevalent Indian dengue strains have not been investigated to date.

Our group has recently published 119 whole-genome dengue sequences from clinical samples across four different sites in India from 2012 to 2018 [39]. This effort has substantially increased the number of whole-genome dengue sequences from India (from 65 to 184 genomes) that now allows careful examination of the evolutionary dynamics of the dengue virus in the country. In this study, we compiled all available whole dengue genomes (n = 184) and E gene (n = 408) sequences to generate the most comprehensive dataset of dengue virus sequences to date from India. We performed spatio-temporal analysis using these sequences to understand the spread of the dengue virus. We also analysed how the virus evolves from its ancestors and in the presence of multiple serotypes. Our work highlights how dengue virus is evolving in India.

## Methods

### Data collection

**Dataset A.** All the published Indian dengue sequences were obtained from the ViPR database [40]. These sequences include both whole-genome sequences and gene segments. Only sequences with location and collection date were used for the analysis (about 88.9% of all sequences). Samples for these sequences were collected between 1956 and 2018 and represent DENV-1 (n = 840), DENV-2 (n = 877), DENV-3 (n = 746) and DENV-4 (n = 179) serotypes.

**Dataset B.**   Global dengue protein-coding sequence records that contain sample collection dates were obtained from the ViPR database [40]. After removal of identical sequences, this dataset included DENV-1 (n = 1800) from 1944–2018, DENV-2 (n = 1395) from 1944–2018, DENV-3 (n = 823) from 1956–2018, DENV-4 (n = 220) from 1956–2018, representing a total of 4238 protein-coding sequences. **Dataset B.1:** For analysis of spatio-temporal dynamics, protein-coding sequences specific to the Indian genotypes were selected from dataset B. We included all unique protein-coding sequences from India and augmented the set with 362 randomly sampled global sequences belonging to the Indian genotypes (S1 Table). This included a total of 522 sequences from DENV-1 genotype I (n = 142), DENV-1 genotype III (n = 93), DENV-2 genotype Cosmopolitan (n = 144), DENV-3 genotype III (n = 96), and DENV-4 genotype I (n = 47).

**Dataset C.**   All Indian E gene amino acid sequences collected since 2000 were obtained from the ViPR database [40] which comprised of DENV-1 (n = 113), DENV-2 (n = 168), DENV-3 (n = 88), and DENV-4 (n = 39) E protein sequences.

State-wise number of dengue cases and deaths over the period 2001 to 2022 was retrieved from https://www.indiastat.com/. Peaks in the number of cases and deaths were assigned for a particular year if the number of cases/deaths in that year increased more than 50% from the previous year. Due to the COVID-19-related disruptions from 2020 onwards, the cases from 2019–2022 were excluded for determining the peaks [41].

## Sequence alignment and phylogenetic analysis

Multiple sequence alignment was performed with protein-coding sequences from dataset B using MUSCLE v3.8.425 [42] implemented in AliView v1.25 [43]. Alignments were manually checked for insertion and deletion errors. Maximum likelihood trees were generated using IQ-TREE v1.6.10 [44] with 1000 bootstraps. A general time-reversible substitution model with unequal base frequency and gamma distribution for rate heterogeneity (GTR+F+I+G4) was selected out of 88 models available using jModelTest [45] implemented in IQ-TREE v1.6.10 [44] based on the Bayesian information criterion. Sylvatic strains EF457905 (DENV-1), EF105379 (DENV-2) and JF262779-80 (DENV-4) were used as outgroups to root the respective trees. The root for DENV-3 was obtained using the best fitting root by the correlation method in TempEst [46]. The maximum likelihood phylogenetic trees were used to obtain the root-to-tip distances using TempEst. Trees were visualized using Figtree v1.4. We assigned new lineages within a genotype if the difference between the sequences from each phylogenetic branch is more than 3% at the nucleotide level and 1% at the amino acid level.

Bayesian analysis was performed with the E gene, NS5 gene and whole-genome sequences from dataset B.1 using the BEAST v1.8.3 [47] to get the substitution rates for each gene. Five sets were generated for each serotype by selecting 80% sequences randomly, the substitution rate was calculated for each run, and the average substitution rate was reported. The constant rate clock model was selected based on the AICM values of the Bayes factor and harmonic means (S2 Table). Markov Chain Monte Carlo (MCMC) was run for $10^7$ generations for each run, and the first 10% of samples were discarded as burn-in. The phylogeographic movement of the virus across the countries was obtained using the Bayesian stochastic search variable selection procedure implemented in BEAST v1.8.3 with whole genome sequences from dataset B.1 (MCMC chain length ~$10^8$). SpreaD3 v0.9.7 [48] was used to visualize the spatio-temporal dynamics of the virus. Tracer v1.6 was used to check the convergence of the chains. Effective sample sizes for the parameters of interest were greater than 200.

Single-likelihood ancestral counting (SLAC) and fixed effects likelihood (FEL) methods were used to identify the positions that are undergoing selection using the HyPhy package

(v2.5.1) [49]. Sites with a *p*-value <0.1 were considered significant only if they were detected by both methods.

## Dynamics of amino acid variation in the envelope gene sequence

Temporal dynamics of E gene amino acid variation was examined in the sequences from South India due to the availability of a larger dataset (n = 164). Sequences with at least 50% coverage (mean coverage of 93%) of the E gene were selected for the analysis. The Hamming distance between the pair of sequences was divided by the length of the overlapping region between the pair. This was further converted to the z-score using the mean and standard deviation to obtain the normalized distance. To extract the dynamics within the serotype, we measured the normalized distance of new variants over the years with respect to the ancestral sequences (sequences from 2007/08 were used as ancestors due to a lack of sufficient sequences before that). The inter-serotype distance was calculated by selecting the sequences from two different serotypes during a particular year. Sample bootstrapping (n = 100) was used to determine the median and standard error of the normalized distances for each year. The time period of oscillations was calculated using an autocorrelation function with different lags for each trace. A lag with the maximum correlation coefficient was assigned as the period of oscillation for that trace. Bootstrap replicates (n = 100) were used to obtain the distribution of the time period. We performed the same analysis with the CprM gene sequences (position 55 to 155 of the coding sequence) from South Indian sequences (n = 411).

Pearson's correlation coefficient was obtained between the serotype and inter-serotype distance dynamics. The robustness of the correlations was checked by three methods. First, by randomly deleting up to 3 data points from each dynamics (1000 bootstraps). Second, by estimating the cross-correlation between individual traces from two comparing groups randomly with 1000 bootstraps. As a control, we also checked correlations between the groups by randomly shuffling the time series of normalized hamming distances (before correlating) to ensure that the correlations do not arise merely from the yearly fluctuations (1000 bootstraps).

## Dominant epitope selection from the database

Experimentally determined epitopes for all dengue serotypes were obtained from the Immunome Browser tool available on the Immune Epitope Database [50]. B cell and T cell epitopes were selected based on their response frequency (RF) score, calculated as reported earlier [51]. Epitopes having RF-score > 0.25 were selected as dominant epitopes for further analysis. For the epitopes examined in multiple studies, the RF-score was calculated by combining the number of subjects from all the studies. Apart from these known epitopes, we also included 77 sites in which residue variation conferred antigenic effects [52].

## Comparison with the vaccine strains

All Indian DENV envelope sequences post-2000 from dataset C were compared with three vaccines: CYD-TDV developed by Sanofi Pasteur, TV003 by NIH/Butantan and TAK-003 by Takeda. Multidimensional scaling based on the Hamming distances between the sequences was used to visualize and evaluate the differences between Indian and vaccine strains.

Homo-dimeric structures of the envelope proteins were obtained by homology modelling using the SWISS-MODEL server [53]. Template PDB structures were selected based on global model quality estimates and QMEAN statistics. Vaccine strain information and template selection for each serotype is shown in the S3 Table. Sites with differences in >10% of sequences were identified as significant differences and were mapped onto the envelope protein structure using PyMOL (v2.4.1) [54].

## Results

### Dengue serotype dynamics in India

The reported dengue cases in 2018 have increased more than 25-fold (three year average) since 2002 in India (S1A Fig). All four geographical regions, namely–North, East, South and West-Central India, show periodic spikes in dengue cases as well as deaths over 2–4 years (S1B–S1D Fig). Comparing all published dengue sequences till 2018 from India (dataset A), we find all four dengue serotypes co-circulating in the country since 2000 (Fig 1). Although dengue sequence reporting from various parts of the country is sporadic and focused on the urban areas, the number of annual cases and deaths in the past two decades correlate well with the number of available sequences each year (S1E Fig, Pearson's correlation coefficient = 0.67). In particular, we noted the increase in DENV-2 and DENV-4 sequences since 2011 and 2014, respectively (Fig 1A). Corresponding to the reported cases, we found periodic peaks in the number of sequences reported from North and South India. Genomes reported from North India show a pattern of serotype replacement in consecutive peaks (Fig 1B). Since most of the sequences from North India were collected from Delhi (~80.5%), the spikes in the dengue sequences in 2006, 2010 and 2013 also correspond well with the outbreaks in Delhi [31,55] (S1C–S1D Fig). Consistent with previous reports, we find DENV-3 dominated during the 2006 outbreak [56] while DENV-1 and DENV-2 serotypes dominated during the outbreaks in 2010 and 2013, respectively [31,57]. Although dengue outbreaks have been observed in East and West-Central India (S1C and S1D Fig), a relatively smaller number of sequences are

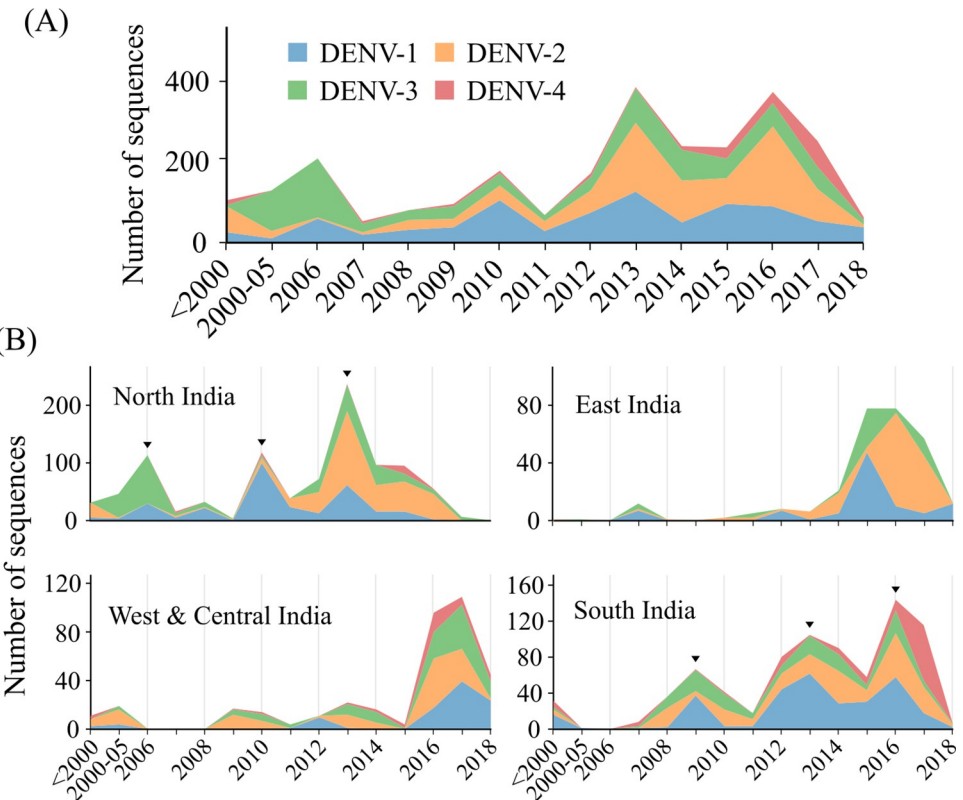

**Fig 1. Year-wise dengue sequences reported from India.** (A) Dengue serotype sequences available from India till 2018. (B) Distribution of dengue serotypes in different regions of India. DENV-1: blue, DENV-2: orange, DENV-3: green, DENV-4: red. Arrows indicate the peaks in the number of sequences.

available from these regions, possibly due to poor genomic surveillance (Fig 1B). Nevertheless, DENV-2 emerged as the dominant serotype in East India in 2016, while all serotypes were identified in West-Central India from 2016 to 2018. In South India, three peaks are evident in the number of sequences corresponding to 2009, 2013 and 2016 (Fig 1B) and correlated peaks in the number of cases/deaths were observed in 2009–10, 2012–13 and 2017 (S1C and S1D Fig). DENV-1 and DENV-2 have been the dominant serotypes over the last decade, but DENV-4 has recently emerged as a major serotype in South India. Similar periodic fluctuations in dengue incidence and replacement of dengue serotypes have been proposed previously to be driven by population-level immunity [18,19], but whether high endemicity and seropositivity are driving case incidences, serotype-specific dynamics and virus evolution in India is not known. In summary, DENV-1 and DENV-3 were the dominating serotypes in India till 2012. DENV-2 has become the dominant serotype in most regions in India since then, and DENV-4 is establishing itself in South India.

## Prevalence and spatio-temporal dynamics of Indian dengue genotypes

Maximum likelihood trees and maximum clade credibility trees of complete genome coding sequences suggest that dengue genotypes are constrained by geography exemplified by the poor intermixing of the genotypes across continents (Figs 2 and S2–S4). Asian sequences consist of two dominant genotypes for all the serotypes, while the recent Indian dengue sequences represent only one of those genotypes (except for DENV-1). Genotype III for DENV-1, cosmopolitan genotype for DENV-2, genotype III for DENV-3, and genotype I for DENV-4 have been the dominating genotypes in the past two decades in India. The Indian genotypes are remarkably divergent from dengue in other regions of the world beyond Asia and merit further examination of their evolutionary dynamics (Fig 2A).

Interestingly, we find distinct temporally and spatially co-circulating lineages within the genotypes in India (Fig 2B–2E), yet it remains unclear what factors contribute to their co-prevalence. Among the two DENV-1 genotypes (I and III), genotype III has been dominant across India (S2 Fig). Within genotype III, two dominant lineages (lineage I and II) are circulating simultaneously within the country. These lineages differ at 39 positions across the genome (S5A Fig) and five positions in the immunologically dominant E gene (T272M, I337F, T339I, V358A and I/V461I). Residues 272 and 337 are on the exposed part of the E protein dimer, and position 272 is linked to having an antigenic effect suggesting differential antigenicity. Even though the phylogenetic clustering did not include the untranslated regions, all the sequences from lineage-I carry a deletion of 21 nucleotides in the hypervariable region of the 3'UTR (nucleotide positions 10294–10314 with respect to the reference sequence NC_001477, S6 Fig). A similar deletion has been reported previously from India [58], but the significance of such a large deletion is unclear. This region is not involved in any stem-loop structures or the genome cyclization region in the 3'UTR. *In vitro* studies have shown that a 19-nucleotide deletion (nucleotide positions 10289–10309) overlapping this region of 3'UTR did not affect the growth kinetics of the virus [59], while larger deletions (nucleotide positions 10274–10728) in the 3'UTR variable region show growth defects in human cell lines [60]. Therefore, it is likely that DENV-1-III lineage-I can tolerate the 3'UTR deletion without a significant fitness cost. On the other hand, a cluster of sequences in DENV-1-III lineage-II carry an insertion of two nucleotides in the hypervariable region of 3'UTR (C10274 and A10297 with respect to NC_001477, S6 Fig) and is restricted to South India. This shows how at least three sets of genotype III strains are co-circulating in the country. Spatio-temporal analysis reveals that DENV-1-III lineage-I is a relatively new lineage that emerged in 1992 (95% HPD: 1990–1995) and is limited to India, Singapore, and China (Figs 2B and S7A and S4 Table). Our analysis shows the

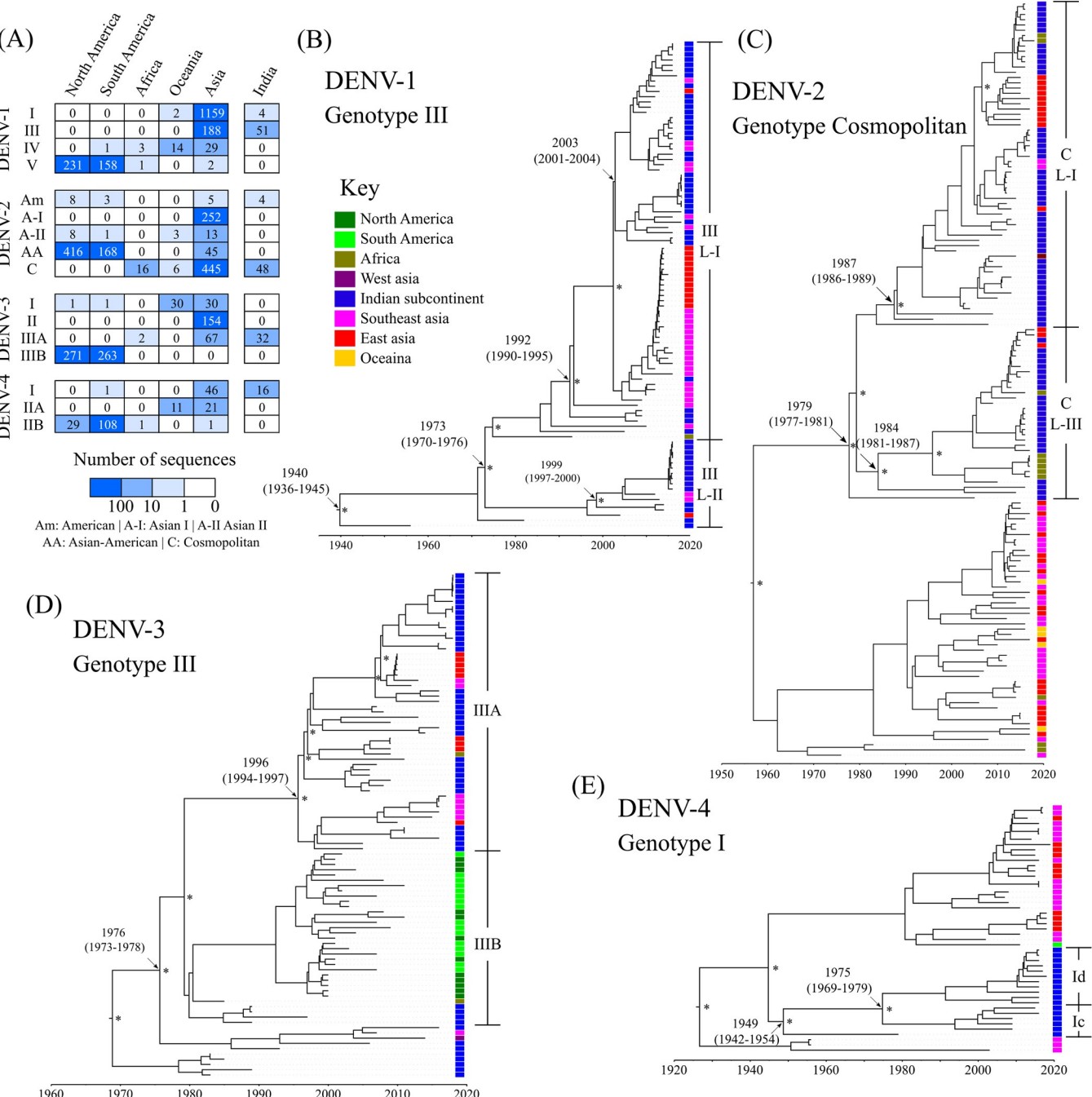

**Fig 2. Dengue genotype distribution and phylogeny.** (A) Heatmap of the number of whole-genome sequences from different geographical regions and India. (B-E) Time-dated phylogenetic trees for all serotypes with the circulating genotypes in India (DENV-1-III, DENV-2-cosmopolitan, DENV-3-III, DENV-4-I). The colour of the rectangle at the tip of the branches represents the region of the sample collection. The estimated time of the common ancestors is denoted for the important nodes along with the 95% highest posterior density (HPD) intervals. Asterisk (*) denotes posterior probability support of ≥ 0.95.

early emergence of this genotype in India, with subsequent spread to South Korea, Comoros, and Singapore between 1990 and 2005 and multiple import-export events between India and Singapore post-2005 (S7A Fig). Apart from this, genotype I of DENV-1, the primary genotype in most Asian countries (Fig 2A), has also been reported exclusively from South India since 2012, suggesting a more recent introduction (S8 Fig).

The earliest DENV-2 sequences from India belong to the American genotype. This genotype was predominant in India before 1971 but was eventually replaced by the cosmopolitan genotype [61]. The DENV-2-cosmopolitan genotype was introduced to India and China in the early 1980s. From here, it has spread to Southeast Asia, Australia, and East Africa (S7B Fig). DENV-2 in India has two distinct co-circulating lineages (lineage-I and III) that emerged in the 80s with a common ancestor that dates back to 1979 (1977–1981) (Figs 2C and S3 and S4 Tables) [32,62]. While prevalent in the Indian subcontinent, cosmopolitan lineage-III has been reported in East Africa, and spatio-temporal analysis suggests that this lineage was most likely exported from India, in line with the study from Kenya [63]. These lineages differ at 41 positions across the genome, including three positions in the envelope gene (S5B Fig), all of which are in known epitopic regions or have antigenic effects (E protein residues 141, 162 and 322), suggesting immune selection pressure playing a role in their divergence. Otherwise, the E gene did not show significantly higher nonsynonymous mutation density compared to other genes (S5E Fig). Similarly, these lineages do not differ in their *in-vitro* virus growth kinetics and disease severity [32]. This could explain how these two lineages have been able to avoid replacement by the other since their emergence.

All the Indian DENV-3 genotype III sequences cluster along with other Asian countries (S4 Fig). DENV-3-III was first detected in Sri Lanka and disseminated to multiple countries over time (S7D Fig). After its initial spread, it evolved into two distinct geographical clusters: lineages IIIA (Asian) and IIIB (American). Most of the spread of genotype IIIB in the Americas was from 1990 to 2005 (S7D Fig). In contrast, diffusion of genotype IIIA in Asia took place after 2005, yet these two lineages have not intermixed (Figs S7D and 2A). In the E gene, these lineages differ at positions 158 and 380, with position 158 being a part of a known B cell epitope. Within lineage IIIA, we found two dominating sub-lineages co-circulating in India (IIIAa and IIIAb). These sub-lineages differ at 26 positions across the genome and eight positions in the E gene (S5C Fig). Although the E gene displays a high density of mutations (S5E Fig), most of these positions (7 of 8) carry the same amino acid in majority of sequences for both the sub-lineages (S5C Fig).

The spatio-temporal reconstruction identifies the earliest sequences of DENV-4-I of Philippines origin (S7C Fig) consistent with the E gene analysis reported earlier [64]. Most of the subsequent spread of DENV-4-I was restricted to Southeast Asian countries. Although it was introduced in India around 1945, there has been a significant increase in reported DENV-4-I sequences from 2015 onwards, especially in South India (Fig 1) [33,65]. It is possible that it remained undetected due to underreporting of mild infections since the primary cases of DENV-4 have been reported to cause mild disease [66,67]. All Indian DENV-4-I sequences cluster separately from the other prevalent genotype I groups (Fig 2E). We identify two lineages in Indian DENV-4 sequences (Ic and Id); DENV-4-Id lineage seems to be replacing DENV-4-Ic since 2016 (Figs 2E and 3C). This novel lineage of DENV-4 (Id) was reported in Pune in 2016 using the CprM sequences [68]. We noticed remarkable differences in the E and NS2A genes between the two lineages (S5D Fig), as discussed in detail in the next section. Overall, all prevalent DENV genotype lineages carry signatures of immune selection contributing to their divergence.

## Emergence of DENV-4-Id suggests dominant immune selection pressure

To examine the selection pressure on the dengue virus in India, we employed FEL (Fixed Effect Likelihood) and SLAC (Single Likelihood Ancestral Counting) methods. Apart from DENV-3, there was no substantial evidence of positive selection, and most amino acid changes were deleterious and negatively selected (S5 and S6 Tables). In DENV-3-III genotype, we

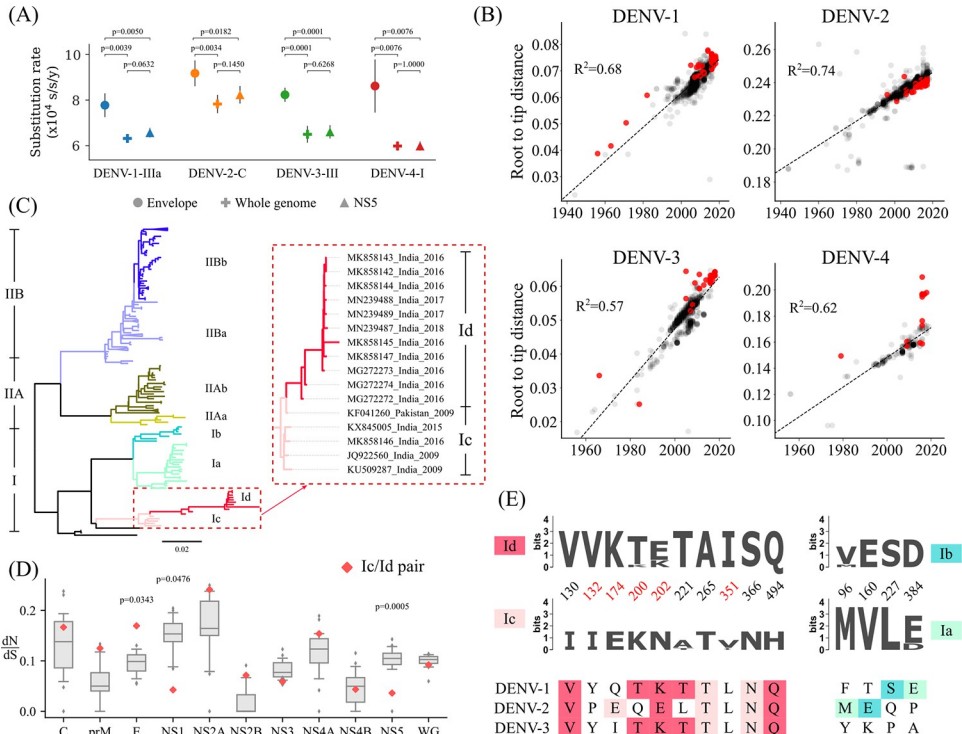

**Fig 3. Substitution rates for Indian genotypes and DENV-4-I lineage.** (A) Comparison of the substitution rates (substitutions per site per year) for different segments of dengue virus for Indian genotypes. The *p*-values were obtained using Welch's t-test (unpaired, two-tailed, unequal variance). (B) Root-to-tip-distance of global dengue whole-genome nucleotide sequences. The Indian sequences are highlighted in red, and the $R^2$ value for the linear fit is shown. (C) The maximum-likelihood phylogenetic tree for complete coding sequences of DENV-4. Distinct branches (corresponding to Ia/b/c/d, IIAa/b and IIBa/b) used for the dN/dS analysis are colour coded. Indian DENV-4 cluster is shown in the inset. (D) dN/dS values for the consensus sequences generated from the branches in (C) are represented as a box plot for each gene. Whiskers denote the 5th and 95th percentile. The dN/dS values for Ic/d pair are marked in red. The generalized extreme Studentized deviate test was performed to find the outliers, and *p*-values are reported. (E) Sequence logo showing the amino acid variations between pairs Ic/d and Ia/b. The amino acids at these locations for DENV-1-3 are shown at the bottom. Highlighted colour depicts the DENV-4 branch with the same amino acid residue.

found a single position in NS5 protein (50I/T) under positive selection, while this site is highly conserved in all other serotypes. Significant negative selection in DENV is consistent with the requirement for horizontal transmission between taxonomically diverse host species, which imposes a strong purifying immune pressure [10,12,13].

Further, the substitution rates of dengue genotypes in India are comparable (7.59E-4, 6.31E-4, 7.83E-4, 6.50E-4 and 6.26E-4 substitutions/site/year for DENV-1-I, DENV-1-III, DENV-2-cosmopolitan, DENV-3-III and DENV-4-I, respectively) (S4 Table) and similar to earlier reported rates [7,62,69]. Although 95% highest posterior density (HPD) intervals for all the genotypes overlap, the largest substitution rate is observed for the DENV-2-cosmopolitan genotype. However, the substitution rates obtained for the E gene were about 26% (11.5% to 44.2%) higher than those for the whole genome (Fig 3A). Interestingly, the substitution rate was 44% larger for the DENV-4-I E gene compared to the whole genome, suggesting high immunological pressure driving the divergence of the DENV-4 E gene. This is consistent with previous reports for the HIV envelope gene and spike protein of the SARS-CoV-2 virus [70,71].

Root-to-tip distance analysis showed that most Indian dengue viruses follow the molecular clock similar to that observed in other parts of the world (Fig 3B). However, Indian DENV-4 lineage Id is highly divergent with residuals larger than three times the interquartile range and displays the most extended branch in the phylogenetic tree (Fig 3B and 3C). Indeed, when we compare dN/dS ratios between branches within the DENV-4 phylogenetic tree (Fig 3C), we found significantly higher dN/dS values for the E gene for the DENV-4-Ic/d pair compared to other DENV-4 clade pairs (Fig 3D). Besides the E gene, NS2A also showed a relatively higher dN/dS ratio and nonsynonymous mutation density (Figs 3D, S5D and S5E). This suggests a role of immune selection pressure in DENV-4-Id evolution consistent with co-evolution of E and NS2A genes correlating with virus antigenicity [52]. On the other hand, NS1 and NS5 genes showed significantly lower dN/dS ratios for the same pair. In fact, 10 out of 34 mutations in DENV-4-Id (with respect to DENV-4-Ic) are present in the E gene (S5D Fig). The E gene variant positions 132, 174, 200, 202 and 351 are also in the known epitopic regions and/or have antigenic effects pointing to the immune escape-driven divergence of DENV-4-Id. Intriguingly, five of the acquired E gene mutations (I130V, K200T, N202K/E, A221T, H494Q) in DENV-4-Id are similar to the DENV-1 and DENV-3 E genes (Fig 3E). This was unexpected since the prior prevalence of DENV-1 and DENV-3 in the country should have forced the DENV-4 to drift away from them. On the other hand, antibody-dependent enhancement may confer some fitness advantage to the DENV-4-Id clade due to shared antigenic features with DENV-1 and DENV-3. Such movement towards other serotypes is not evident in the phylogenetically related DENV-4-Ia/b pair (Fig 3E), suggesting a unique signature of the Indian DENV-4-Id lineage.

## Dynamics of E gene evolution displays recurring variation

Taking a cue from the divergence of DENV-4-Id, we examined whether the high seroprevalence can play a role in the evolution of dengue in India. Dengue cross-reactive immunity has been shown to shape the antigenic evolution of dengue for the E gene at a city level [11]. Since serotype replacements in the different regions of the country display temporal fluctuations (Fig 1C), we asked whether there are signatures of immunity-driven evolution of dengue serotypes across large endemic areas. In the absence of antigenicity data, we evaluated the variation in amino acid sequences of the E gene longitudinally. To validate this approach, we rely on neutralization antibody titers of sera collected from humans (post-monovalent vaccination) and African green monkeys (post-infection) tested against a panel of viruses [72]. We found general correlation between the antibody titers and the similarity between the E protein sequences with the two datasets of antibody titer data (Pearson's correlation coefficient: 0.530, S9 Fig). In South India, we find that, in general, the E gene diverges from the ancestral sequence for all serotypes, but this divergence fluctuates over time (Fig 4A). Overall, in our dataset, the E gene sequences drift away from their respective ancestral sequences, evolve to be similar and then diverge repeatedly. This behaviour was pronounced in DENV-2-4 with an estimated time period (peak-to-peak) of about three years (2.92 ± 0.58 years for DENV-2, 3.55 ± 0.9 years for DENV-3 and 2.99 ± 0.64 years for DENV-4). While we could not estimate a time period for a similar E gene dynamic in the case of DENV-1, we did note a peak in divergence in 2012–13 arising from a singular genotype I outbreak during a period of genotype III prevalence.

Interestingly, the observed dynamics of the E gene amino acid variations were also correlated among the serotypes (Figs 4C and S10). The high correlation suggests that the divergence in each serotype was synchronous, i.e., when DENV-1 drift away (or converges) from its ancestral sequences, a similar dynamics is observed for DENV-2-4 with respect to their

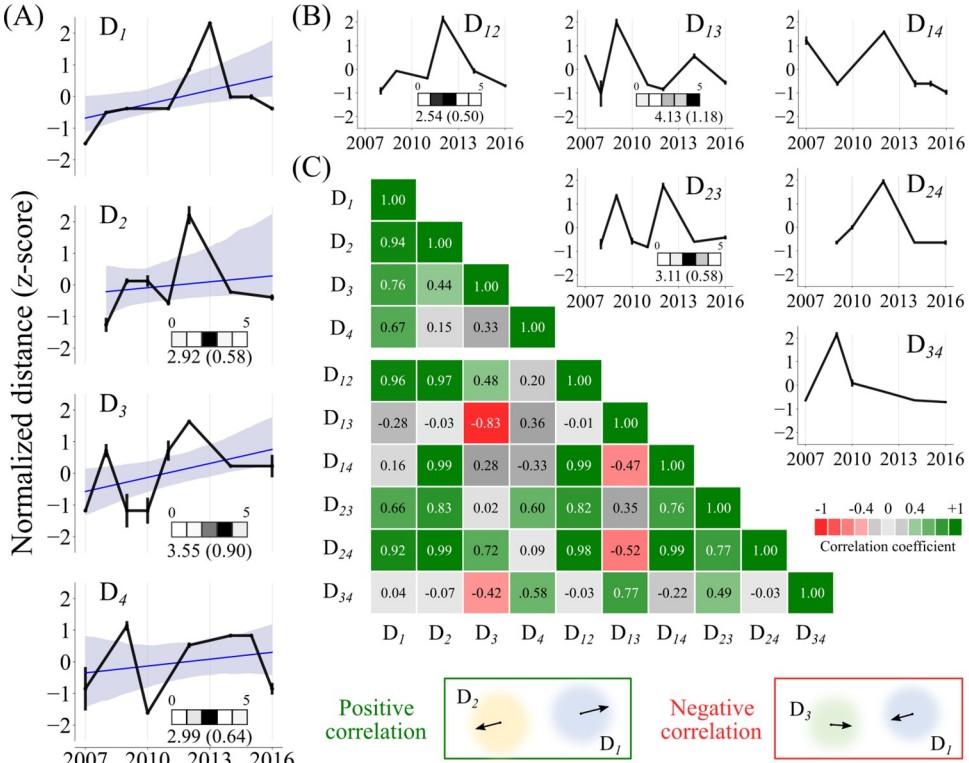

**Fig 4. Dynamics of E gene amino acid variation in South India between 2007 and 2016.** (A) Normalized amino acid distances with respect to the 2007/2008 dengue strains is plotted in black. Hamming distances were normalized with the sequence length and converted to a z-score. Distances were bootstrap sampled (n = 100) to calculate the reported median. Error bars represent standard error. The blue line represents a linear regression fit to the data with 80% (gray shade) confidence interval. $D_i$ indicates the distance dynamics between DENV-i from its (2007/2008) ancestral sequence. (B) Year-wise normalised interserotype distances (as z-score) calculated by randomly selecting the sequences from each serotype every year (100 bootstraps) is plotted. The inset heatmap depicts the time period of oscillation binned yearly (range of 0 to 5 years). The mean and standard deviation (in brackets) of the time period distribution is denoted next to the heatmap. $D_{ij}$ indicates the interserotype distance dynamics between $i$-th and $j$-th serotype. (C) Median of the Pearson's correlation coefficients between the distance dynamics $D_i$ and $D_{ij}$ is shown as a heatmap with corresponding colormap. A schematic depicting the relative evolution of dengue serotypes in two-dimensional sequence space with positive and negative correlation between $D_{ij}$ and $D_i$ or $D_j$ is shown at the bottom. Central point represents the ancestral strain, and the arrow represents the direction of the virus evolution.

ancestral sequences (Figs 4C and S10). To understand whether the evolution of the dengue virus is shaped by the interserotype cross-reactive immunity in the population, we examined the dynamics of interserotype distance $D_{ij}$ (year-wise hamming distance between DENV-$i$ and DENV-$j$) (Fig 4B). We observe that the interserotype distances also displayed fluctuations over a similar time period of 2–4 years (2.54 ± 0.5 years for $D_{12}$, 4.13 ± 1.18 years for $D_{13}$ and 3.11 ± 0.58 years for $D_{23}$), suggesting an interplay between the serotypes at the population level. We also checked whether the serotype sequences became similar to each other or drifted apart based on the correlations between the serotype and interserotype evolutionary dynamics. For instance, when DENV-1 and DENV-2 display divergence (or convergence) with respect to their ancestral sequences (Fig 4A), the distance between the DENV-1 and DENV-2 ($D_{12}$) also increase (or decrease). While DENV-1 dynamics did not correlate with $D_{13}$ and $D_{14}$ dynamics, DENV-2 strongly correlated with all other interserotypic dynamics. In contrast, DENV-3 showed a negative correlation with $D_{13}$ and $D_{34}$ dynamics suggesting that as DENV-3 diverges, it moves closer to the circulating DENV-1 and DENV-4 strains (Figs 4C and S10). These

signatures are specific to the immunologically dominant E gene. For example, when we compare the CprM gene sequences from South India, the fluctuations were not significant. When CprM Hamming distance fluctuations are observed as in the case of $D_1$ dynamics, it is linked to the transient introduction of genotype I in 2012 (S11A Fig). Consistent with this, we found no significant correlations between the within or inter-serotypic viral dynamics, except for DENV-4 (S11B and S11C Fig). Due to the availability of only 21 sequences for DENV-4 (and n = 3 between 2008–2015), we do not consider these correlations as significant.

We interpret these coupled fluctuations between dengue serotypes in light of population-level cross-reactive immunity and antibody-dependent enhancement. When homotypic immunity is present in the population, the serotypes drift apart, manifesting in positive correlations between serotype divergence and interserotype dynamics, as observed for DENV-1 and DENV-2 (Fig 4C). However, when population immunity is poor against a particular serotype (as with a new introduction or waning levels), similarity to that serotype confers an advantage due to the presence of cross-reactive antibodies and associated ADE [14,15]. This is consistent with the negative correlation observed between DENV-3 and interserotype dynamics $D_{13}$ and $D_{34}$ with dropping DENV-3 prevalence and its replacement by DENV-1 and DENV-4 in South India (Figs 4C and 1B). Indeed, recent reports have argued the interserotypic convergence of dengue antigenicity to be correlated to the outbreaks [11].

## Divergence of Indian dengue virus from vaccine strains

All the vaccine strains used by tetravalent vaccines are based on the strains isolated between 1964 and 1988. As the prevalent Indian dengue viruses are evolving under high population seropositivity, they might be diverging antigenically from the vaccine strains as well. Additionally, genotypes used in the vaccine strains are not observed in India (S3 Table). Therefore, we examined the E protein differences between the vaccine and the circulating dengue virus in India. Dimensionality reduction analysis of amino acid differences between the virus strains shows that dengue E gene sequences from India cluster together but are removed from the vaccine strains for all serotypes (Fig 5A). Only the DENV-1 and DENV-2 strains of CYD-TDV are close to the DENV-1-I and DENV-2-cosmopolitan clusters present in India. More than half of the significant amino acid differences (50% DENV-1, 68% DENV-2, 92% DENV-3 and 50% DENV-4) lie on the exposed surface of the envelope protein on the virus, which is accessible to the majority of the antibodies (Figs 5B and S12 and S7 Table). Overall, ~ 6% (2.7–13.5%) of all known epitopic regions are different in Indian dengue sequences compared to the vaccine strains (S8 Table). Further, almost half (34.6–66.7%) of all the E protein variations lie either in known epitopic regions selected from the IEDB database or have a positive antigenic effect [52] (S7 Table). This is in line with our assertion that the dengue virus in India has evolved under immunological selection pressures. Further investigation is needed to assess the impact of these mutations on vaccine efficacy in India.

Indian variants of DENV-1 and DENV-4 are distinct from all the vaccines compared to DENV-2 and DENV-3 (Fig 5C). Among the E protein domains, EDIII, a known antigenic domain and target of recent Indian dengue vaccine efforts [39,73], carry the largest density of variations, further confirming the immune escape-driven evolution of the protein. Apart from EDIII, the transmembrane region of the vaccines (especially for TAK-003) is distinctly different from the Indian variants (S7 Table). This region has been reported to be highly antigenic in DENV-2 [74,75]. Consistent with the role of the stem region in the viral membrane fusion process, it is highly conserved. Therefore, antigenic differences of Indian dengue viruses with respect to the vaccines define the majority of E protein differences. This can have important implications for vaccine efficacy, where vaccines will likely evoke poor

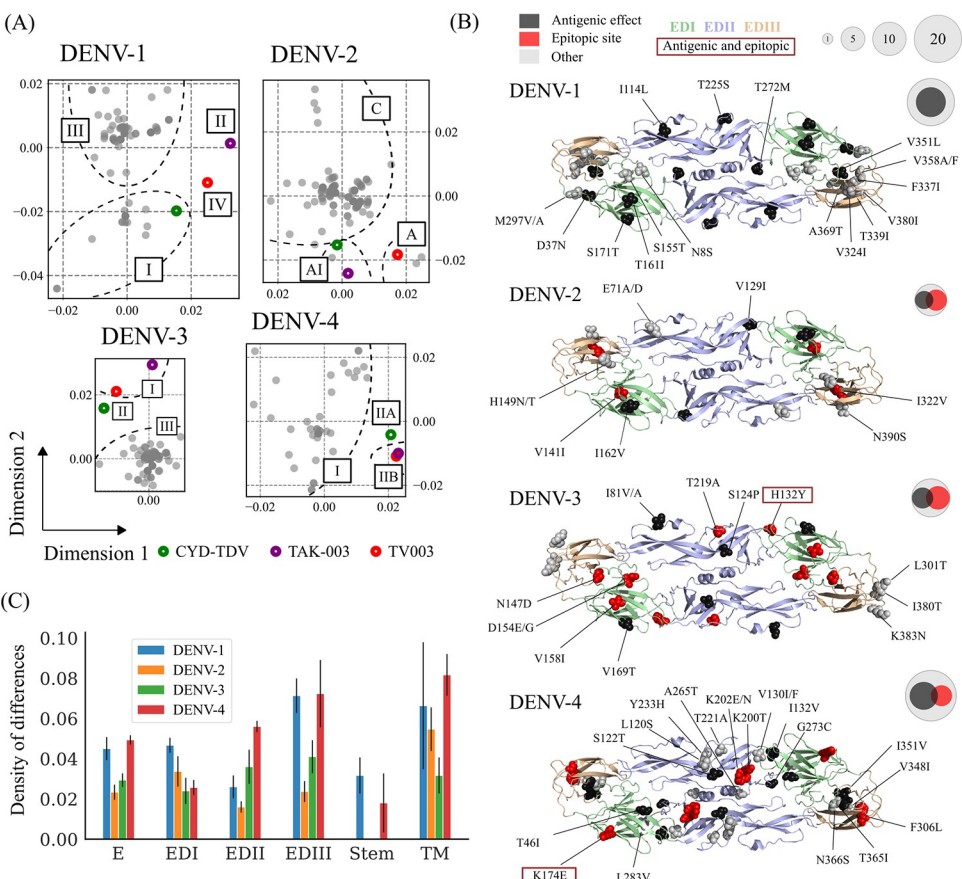

**Fig 5. Comparison of Indian envelope protein sequences with vaccine strains.** (A) Relative amino acid differences between Indian envelope sequences (post-2000) and vaccine strains (CYD-TDV, TV003 and TAK-003). The multidimensional scaling was used to reduce the dimensions of the data using the hamming distance matrix. Genotype cluster boundaries are indicated with dotted lines for visual clarity. Indian sequences are shown in grey; vaccine strains are shown with empty circles. (B) Amino acid differences at a frequency >10% with respect to the CYD-TDV vaccine are shown on the envelope protein dimer structure. Different positions in the known epitopic regions are shown in red. Variable residues within the predicted antigenic effect positions are shown in black. Residues present in epitopic regions with predicted antigenic effects are highlighted with a box. Venn diagram shows the total number of differences in the full E protein (including the stem and transmembrane region), epitopic sites and antigenic sites. The circle size represents the number of differences category-wise. (C) The density of differences (number of differences/ length of the domain) is shown for different domains of the envelope protein (envelope domains EDI-III, stem and transmembrane (TM) region). The error bars represent the standard deviation across the three vaccine strains.

neutralization against dengue virus in India. For example, monoclonal antibodies targeting the EDIII region of dengue display a 100-fold variation in neutralization titer across different genotypes [76,77]. Similarly, challenge studies (in humans, ClinicalTrials.gov Identifier: NCT03416036 and Cynomolgus Macaques) show complete protection against vaccine genotypes but confer only partial protection against other genotypes [78,79]. For the DENV-4-II based CYD-TDV vaccine, efficacy in young children against DENV-4-I is only 23.9%, which has been attributed to eight specific residues in the E/prM gene [38]. Four of these (T46I, L120S, F461L, and T478S) dramatically reduce vaccine efficacy (from as high as ~75% to less than 20%) [38]. DENV-4 in TV003 and TAK-003 vaccines also share three of these residue variations, implying potentially lower vaccine efficacy against the DENV-4-I genotype prevalent in India.

## Discussion

Due to the immense public health burden from dengue infections in India, it is important to understand the genetic diversity, spatial incidence, effectiveness of vaccines and the potential emergence of new variants of the dengue virus in the region. It is expected that a combination of high seroprevalence and co-circulation of all dengue serotypes shapes the evolution of the dengue virus in the country, but the outcome of the interplay between direct and indirect factors has been unclear. Using the demonstrated concordance between E gene divergence and antigenicity of dengue virus (S9 Fig), we probe the evolutionary dynamics of the virus in India. Consistent with immune evasion, we find that the highly immunogenic dengue E gene gradually diverges for all serotypes from their ancestral sequences over time. However, these E gene differences within each serotype were superimposed with recurrent fluctuations with a period of about three years. It is possible that opposing evolutionary constraints imparted by immune selection on one hand and E protein functional fitness on the other could manifest such fluctuations. Similar fluctuations in dengue antigenicity have been reported from Bangkok, Thailand [11], suggesting that immune selection pressure is a prominent driver for such fluctuations during virus divergence in endemic regions.

Interestingly, dengue virus interserotypic evolution is also intricately coupled to each other and displays temporally correlated fluctuations. We speculate that the correlation between the intraserotype and interserotype evolutionary dynamics arises from the interplay of multiple serotypes with the pre-existing heterotypic immunity levels in India (Fig 6). Among the serotypes, antigenically related serotypes display larger cross-reactivity. The level of pre-existing cross-reactive antibodies can modulate the viral load and severity of the disease during the secondary infection [15,16]. Despite long-term protection from homotypic secondary infection, protection from reinfection with the other serotypes is transient (Fig 6A). As this cross-reactive immunity wanes, the chance of ADE increases at intermediate levels of the antibodies [15]. With further reduction in antibody titers, antibody-mediated enhancement or protection no longer contributes to the risk of infection. Therefore, high antibody titers confer protection, but intermediate levels can increase viral load and severity [15,16]. For closely related serotypes, more cross-reactive antibodies can lead to a broader ADE window without effective neutralization (Fig 6B). This can contribute to a larger viral load, longer duration of infection, higher risk of transmission and increased severity. Thus, pre-existing immunity can provide an evolutionary advantage to an antigenically 'similar' virus and promote convergence towards related serotypes. This could contribute to the co-evolution of 'antigenic cousins' as detected in our analysis of Indian dengue genotypes. When this selection pressure no longer constrains the virus (e.g., due to a reduction in cross-reactive antibodies), it is free to diverge away from the ancestral strains. Comparable time scales for observed E gene variations and the decay dynamics of interserotypic cross-reactive antibodies [22–24] suggest that such an interplay between the immune selection of dengue serotypes might be at play.

Another implication of waning protection against heterotypic infection and robust homotypic immunity contributes to serotype replacement events [19–21]. We speculate that the serotype with the cross-reactive antibody levels that manifest in strong ADE effects would have a distinctive advantage in secondary infections. This can explain how major outbreaks coincide with serotype or lineage shifts [20,21]. It also explains the close match between cross-reactive protection decay and the time interval between the heterotypic outbreak cycles in India (Figs 1 and S1). Understanding the interplay of population-level immunity with dengue virus evolutionary dynamics may allow the prediction of future outbreak serotypes and genotypes.

In South India, DENV-1/2/3 were dominating serotypes till 2015, and DENV-4 emerged as one of the dominant serotypes in South India thereafter (Fig 1B) [33,65]. Most of the

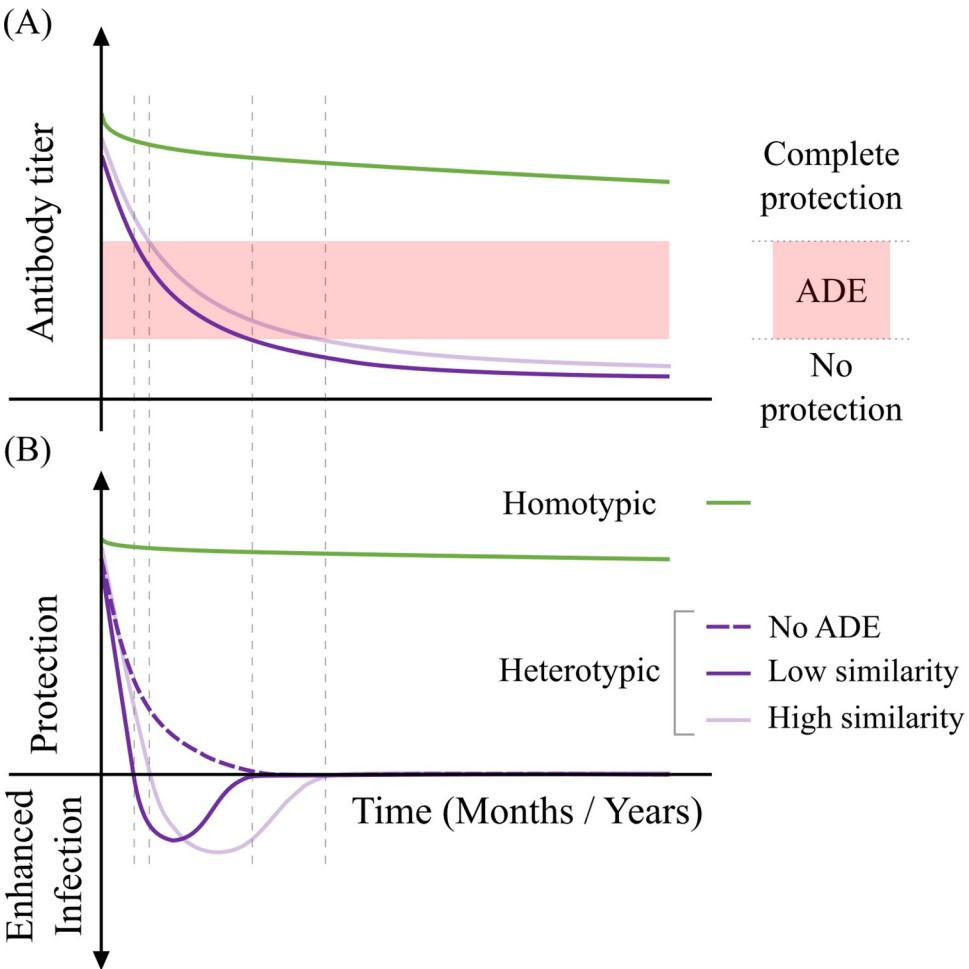

**Fig 6. Schematic for antibody-mediated protection from secondary dengue infection.** (A) Antibody decay dynamics and (B) corresponding level of protection during the secondary infection. Homotypic antibodies and protection from homotypic infection wane slowly (green). Cross-reactive antibodies to heterotypic dengue and protection wane faster (purple). Sequence similarity between the primary and secondary infecting serotypes is shown by shades of purple. The red shaded area represents the window of the antibody titer in which ADE is in effect. The dotted purple line represents protection response in the absence of ADE.

divergence in the Indian DENV-4-Id lineage is largely restricted to immunodominant E and NS2A proteins, again pointing to immune evasion driving its emergence. Consistent with our observation of interserotype correlations, we observed this lineage is moving towards DENV-1 and DENV-3 serotypes. Prior seropositivity due to exposure to DENV-1/2/3 prevents new outbreaks by these serotypes but enhanced similarity of DENV-4-Id to these serotypes might contribute to ADE mediated increased transmission during the secondary infection with DENV-4 in the Indian population. Although DENV-4 was traditionally considered less virulent, it has previously replaced circulating serotypes and caused epidemics in the Pacific region [80,81]. More interestingly, a strong association of DENV-4 with secondary infections (97%) observed in Thailand suggests that DENV-4 can co-opt immunity-driven factors to overcome fitness limitations [82]. High seroprevalence and identification of the fast evolving 'antigenically related' DENV-4 genotype in South India augurs an increased frequency of DENV-4 outbreaks in other parts of the country and possible association with increased severity.

Phylogenetics analysis using the global dengue virus sequences highlights the geographical restriction of dengue viral diversity (Figs 2 and S7). Despite globalization and increased human mobility, dengue genotypes are contained within broad geographical boundaries, albeit with prominent intermixing between neighbouring countries [83]. This could arise from the limited lifecycle of the virus in the host and vector, climate and biodiversity. As a consequence, Indian genotypes are highly divergent from the genotypes in other continents as well as the current vaccine strains. Moreover, most of the variations from vaccine strains are concentrated on the exposed regions of the E protein, including the EDIII region, which plays a significant role in defining the antigenicity [84] and is the primary target of the neutralizing antibodies [85,86]. Therefore, an in-depth characterization of the effect of the reported differences on the antibody titers is required to assess the efficacy of prospective dengue vaccines for India. One can argue that regionally tailored vaccines would be more efficacious due to the geographical restriction of dengue [83]. However, the impact of immunity developed by vaccination on the future course of dengue virus evolution needs to be considered carefully.

The evolution of dengue serotypes (and other flaviviruses [87]) mediated by heterotypic immunity argues that their dynamics is intertwined that cannot be inferred fully by studying them individually. Incorporating such cross-serotype virus-host immunological interactions can improve dengue epidemiological modelling in the future. As the incidences of dengue virus continue to increase worldwide, the host immunity-driven virus evolution would need to be considered carefully to devise interventions.

## Supporting information

**S1 Fig. Dengue cases and deaths reported in India.** (A) Yearly dengue cases and deaths reported in India from 2001 to 2022 (B) Map of India denoting different regions (NI: North, EI: East, WCI: West & Central and SI: South India). The base map was obtained from https://mapsvg.com/maps/india, licensed under CC BY 4.0. The map was modified to show the region's boundaries. (C) Yearly dengue cases and (D) deaths reported in different regions from 2001 to 2022. Data for states is colour coded in each plot. Black arrows denote peaks in the number of cases/death. The average time period between two consecutive peaks is reported for each case. Data from 2019–2022 was excluded while considering the peaks due to the effect of COVID-19-related disruptions. (E) Scatter plot showing the association between the number of dengue sequences from India and cases/ deaths reported. Pearson's correlation coefficient is reported.
(PNG)

**S2 Fig. Maximum-likelihood phylogenetic tree for DENV-1.** All available dated complete coding nucleotide sequences from the human infections were used for tree construction (n = 1800). Branches are collapsed to aid visualization. The detailed tree structure is shown for the neighbouring sequences in the insets. Grey shades within each tree show different genotypes, as indicated on the left side of each tree. The green regions carry the Indian sequences within the tree. The scale for the branch lengths is shown at the bottom of each tree.
(PNG)

**S3 Fig. Maximum-likelihood phylogenetic tree for DENV-2.** All available dated complete coding nucleotide sequences from the human host were used for tree construction (n = 1395). Branches are collapsed to aid visualization. The detailed tree structure is shown for the neighbouring sequences in the insets. Grey shades within each tree show different genotypes, as indicated on the left side of each tree. The blue regions show Indian sequences within the tree.

The scale for the branch lengths is shown at the bottom of each tree.
(PNG)

**S4 Fig. Maximum-likelihood phylogenetic tree for DENV-3.** All available dated complete coding nucleotide sequences from the human host were used for tree construction (n = 823). Branches are collapsed to aid visualization. The detailed tree structure is shown for the neighbouring sequences in the insets. Grey shades within each tree show different genotypes, as indicated on the left side of each tree. The yellow regions highlight clades with Indian sequences within the tree. The scale for the branch lengths is shown at the bottom of each tree.
(PNG)

**S5 Fig. Difference among Indian lineages/ sub-lineages.** (A-D) Amino acid differences between the Indian lineages and sub-lineages are depicted using the sequence logo for each serotype with residue position in the dengue polyprotein listed in between. Gene boundaries are denoted on top, and amino acid positions within the E gene are shown below. The epitopic residue positions in the E gene are highlighted in red. (E) Association between the number of non-synonymous mutations and gene length for all the lineage/ sub-lineage pairs, red lines indicate 95% confidence interval for the regression line. The outlier points (in red) are denoted with the gene name.
(PNG)

**S6 Fig. Multiple sequence alignment of Indian DENV-1-III sequences and the reference sequence (NC001477) in the variable region.** A deletion of 21 nucleotides in the variable region of 3'UTR of DENV-1-III lineage-I sequences is noted. Deleted nucleotide positions 10294–10314 with respect to the reference sequence are shown in the red box. Insertion of 2 nucleotides: C and A, at 10274 and 10297 in DENV-1-III lineage-II sequences, is shown in the yellow boxes.
(PNG)

**S7 Fig. Spatio-temporal dynamics of DENV-1-4 (Indian genotypes).** The movement of the dengue virus within different countries is shown for all serotypes. (A-D) Countries involved in the spread are shown in dark grey. Arrow lines represent the probable direction of the spread. The predicted time of virus introduction to the country is colour coded and shown at the bottom. The base map of the world was obtained from https://mapsvg.com/maps/world, licensed under CC BY 4.0.
(PNG)

**S8 Fig. Time-dated phylogenetic tree for DENV-1-I.** Time-dated trees for DENV-1-I. The x-axis denotes the time in years. The time of the year at important nodes is shown along with the 95% HPD. The posterior probability is shown at each node.
(PNG)

**S9 Fig. Sequence similarity and neutralization antibody titer.** Correlation between neutralization antibody titer collected from humans (x) collected post-vaccination and African green monkeys (o) collected post-infection and tested against a panel of viruses. The X-axis denotes the percentage similarity between the amino acid sequences of the infecting and test viruses.
(PNG)

**S10 Fig. Distribution of correlation coefficients between within-serotype dynamics and inter-serotype distance dynamics.** (A) Distributions were obtained by deleting up to 3 data points randomly from individual dynamics (1000 bootstraps). (B) Histograms of correlation coefficients were obtained by randomly selecting the individual traces (1000 bootstraps). (C)

Histogram of correlation coefficients obtained by shuffling the time points randomly (1000 bootstraps). Green, red and grey indicate positive ($\rho > 0.4$), negative ($\rho < -0.4$), and weak correlation.
(PNG)

**S11 Fig. Dynamics of amino acid variation in the CprM gene sequences in South India isolated between 2007 and 2016.** (A) Normalized amino acid distances with respect to the 2007/2008 dengue strains is plotted in black. Hamming distances were normalized with the sequence length and converted to a z-score. Distances were bootstrap sampled (n = 100) to calculate the reported median. Error bars represent standard error. A line and grey shaded region represent a linear regression line with an 80% confidence interval. $D_i$ indicates the distance dynamics between DENV-i from its (2007/2008) ancestral sequence. (B) Year-wise normalised interserotype distances (as z-score) are reported as calculated by randomly selecting the sequences from each serotype every year (100 bootstraps) is plotted. $D_{ij}$ indicates the interserotype distance dynamics between $i$-th and $j$-th serotype. (C) Distribution of correlation coefficients for within-serotype dynamics and inter-serotype distance dynamics. Histograms were obtained by deleting up to 3 data points randomly (1000 bootstraps). Correlations involving the DENV-4 serotype are highlighted with yellow background and ignored due to the lack of enough sequences. Green and red indicate positive ($\rho > 0.4$) and negative correlation ($\rho < -0.4$), respectively, while grey indicates weak or no correlation.
(PNG)

**S12 Fig. Comparison of Indian envelope protein sequences with vaccine strains (TV003 & TAK-003).** Amino acid differences at a frequency >10% with respect to (A) TV003 and (B) TAK-003 vaccine are shown on the envelope protein dimer structure in grey, red and black. Different positions in the known epitopic regions are shown in red. Positions in the predicted antigenic effect positions are shown in black. Residues present in epitopic regions with predicted antigenic effects are highlighted with a box. Venn diagram shows the total number of differences in the full E protein (including the stem and transmembrane region), epitopic sites and antigenic sites. The circle size represents the number of differences category-wise.
(PNG)

**S1 Table. Accession Numbers for all datasets.** A list of accession numbers for Dataset B, B.1 and C is given. Dataset B.1 also includes the place and year of collection of the corresponding dengue sequencing sample.
(XLSX)

**S2 Table. Model selection for Bayesian analysis.** The clock models for running the BEAST analysis were selected based on (A) Akaike information criterion and (B) harmonic mean method. Lower AICM values and higher HM values denote a better model. Optimum values and models are shown in bold. (C) Chain lengths used BEAST MCMC analysis for each serotype for phylogeographic analysis.
(XLSX)

**S3 Table. Virus strains used in the vaccine.** Strains used for CYD-TDV, TV003 and TAK-003 vaccines along with their genotype, country of origin and collection year. Also, the template PDB structures used for homology modelling for each of the vaccines and the respective genotype is listed.
(XLSX)

**S4 Table. Substitution rates and time to most recent common ancestors.** Substitution rates obtained from Bayesian MCMC analysis. The average substitution rate obtained by running

BEAST analysis five times with 80% sequences selected randomly is reported. Rates are expressed in substitutions per site per year (s/s/y). Time to most recent common ancestors (TMRCA) is reported for recent Indian sequences along with the 95% highest posterior density (HPD) intervals.
(XLSX)

**S5 Table. Selection pressure analysis summary.** The number of sites under positive ($N_{pos}$) and negative selection ($N_{neg}$) with different datasets using the fixed-effect likelihood method (FEL) and single likelihood ancestor counting method (SLAC).
(XLSX)

**S6 Table. Selection pressure analysis.** Sites under positive and negative selection based on the *p*-values < 0.1 using the fixed-effect likelihood method (FEL) and single likelihood ancestor counting (SLAC) method with different datasets.
(XLSX)

**S7 Table. Amino acid differences in Indian DENV envelope gene sequences compared to the vaccine strains.** Amino acid differences in at least 10% of the samples with respect to the CYD-TDV, TV003 and TAK-003 vaccine strains. The sites that fall within the known B or T cell epitopes are highlighted in red. Sites with antigenic effects are highlighted in grey.
(XLSX)

**S8 Table. Percentage of variant sites out of all the known antigenic sites.** Percentage of sites showing the amino acid differences in Indian strains out of all the known antigenic sites.
(XLSX)

## Acknowledgments

We would like to thank Mary Dias, Guruprasad Madigeshi, Jayanthi Shastri, Ravisekhar Gadepalli, and Saranya Marimuthu for their help and valuable feedback on this work.

## Author Contributions

**Conceptualization:** Suraj Jagtap, Chitra Pattabiraman, Rahul Roy.

**Data curation:** Suraj Jagtap, Chitra Pattabiraman, Arun Sankaradoss.

**Formal analysis:** Suraj Jagtap, Chitra Pattabiraman.

**Funding acquisition:** Sudhir Krishna, Rahul Roy.

**Investigation:** Suraj Jagtap, Chitra Pattabiraman.

**Methodology:** Suraj Jagtap, Chitra Pattabiraman.

**Project administration:** Rahul Roy.

**Software:** Suraj Jagtap.

**Supervision:** Sudhir Krishna, Rahul Roy.

**Validation:** Suraj Jagtap, Chitra Pattabiraman, Rahul Roy.

**Visualization:** Suraj Jagtap.

**Writing – original draft:** Suraj Jagtap, Rahul Roy.

**Writing – review & editing:** Suraj Jagtap, Chitra Pattabiraman, Arun Sankaradoss, Sudhir Krishna, Rahul Roy.

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
