## [Decision Letter · Decision Letter 0]

15 Nov 2022

Dear Dr. Roy,

Thank you very much for submitting your manuscript "Evolutionary dynamics of dengue virus in India" for consideration at PLOS Pathogens. As with all papers reviewed by the journal, your manuscript was reviewed by members of the editorial board and by several independent reviewers. In light of the reviews (below this email), we would like to invite the resubmission of a significantly-revised version that takes into account the reviewers' comments.

Most of the reviewers indicate that this study, a detailed evaluation of dengue virus genetic diversity in India, addresses an important gap in the literature. However, all reviewers were concerned about over-interpretation of results, especially given limitations in antigenic data and other epidemiological data. Addressing the reviewers' comments about areas of the manuscript that are unclear, toning down interpretations beyond what is supported by the data, clearly stating limitations, and including additional methodological details and phylogenetic analyses as suggested should be addressed prior to publication. Reviewer 2 also proposed including clinical and epidemiological data to support the genetic analyses. We understand inclusion of such datasets may not be possible. If the other reviewer comments and concerns about the text and additional analyses can be addressed, this would make the manuscript suitable for publication.

We cannot make any decision about publication until we have seen the revised manuscript and your response to the reviewers' comments. Your revised manuscript is also likely to be sent to reviewers for further evaluation.

Sincerely,

Leah Katzelnick

Guest Editor

PLOS Pathogens

Sonja Best

Section Editor

PLOS Pathogens

Kasturi Haldar

Editor-in-Chief

PLOS Pathogens

orcid.org/0000-0001-5065-158X

Michael Malim

Editor-in-Chief

PLOS Pathogens

orcid.org/0000-0002-7699-2064

Reviewer's Responses to Questions

**Part I - Summary**

Reviewer #1: This is an important study considering the genetic diversity of dengue in India, an area about which there is little known for dengue. I found some of the manuscript a little confusing, and required some further clarifications on the methods and on the reasoning for some of the conclusions drawn.

Reviewer #2: The manuscript describes the evolutionary dynamics of dengue virus in India. Overall, the manuscript sufficiently reports finding and provides thorough discussion. However, there are some limitations regarding the general findings, in which no specific or breakthrough findings were described. The cyclical patterns of dengue virus serotype is a common phenomenon in all endemic countries. The genotypes co-circulation and replacement is also commonly observed in other countries.

Moreover, although the methods and analyses were sufficient, the sample number is probably not strong enough to be representative of the whole India, especially because there are 4 DENV serotypes present that need to be equally represented. Authors may try to explain the sample size in term of their representative-ness in the manuscript, both serotypically and geographically. Given the massive population of India, the authors must be able to convince the reader that the sample size is powerful enough to draw conclusion.

The manuscript may be more useful if it includes the clinical and epidemiological data so that the viral genetic data will be more meaningful and may be used to mitigate future epidemics. This, and other limitations, are not specifically described in the manuscript.

Reviewer #3: In this paper, the authors explore the genetic diversity of dengue viruses in India. They compare the sequences of the circulating viruses with that of vaccines, demonstrating that they are quite different in antigenically relevant sites. India probably suffers from the greatest burden of dengue infection worldwide, but there have been few studies that have characterised the genetic diversity in the country. This study is therefore welcome. I have a few comments.

**Part III – Minor Issues: Editorial and Data Presentation Modifications**

Reviewer #1: The latter section of the introduction appears to give a summary of all the results. I would suggest removing this section from here and putting in the discussion. Some of the comments below are on this section and these will hold when the section is moved to discussion.

Line 84-85: What do you mean by serotype replacement?

Could you be more specific about how ADE plays a role in increasing dengue infections (eg increasing from what, and is it infections or cases?)

Line 114: I assume other sequences have been brought in for this part of the analysis. Please give a bit more detail on where this result is from.

Line 116-118: What is the interpretation of the result about Hamming distances? Is this linked to the comment about cross reactive immunity? It currently isn’t clear- suggest rephrasing.

Line 120: What is the “this” that has lead to the highly divergent DENV4 and what is the evidence that “this” has caused it? Why just for DENV 4?

Figure 1: Are there any differences between the two directions of the y axis 0-100? Suggest updating legend to explain or editing figure so that is is more easily interpretable.

Subset C: It isn’t clear from the description if this dataset is sequences from India, or all sequences of those found in India (but including elsewhere). Suggest rephrasing.

Why was 2019 removed for COVID?

Line 236-237 and Line 242- 243: Appears to be a repetition of the DENV4 emergence in South Indian result.

Line 253 -255: Spatially within India or globally

For the global comparisons it would be helpful to have some idea of the numbers of sequence available for different countries. It is currently hard to interpret whether these genotypes are known to not circulate in some countries or if there just isn’t much data.

It would be helpful to state what is the evidence for the temporality inferred from the trees?

Line 418-419: Please cite a reference for these regions with variations as having a positive antigenic impact or epitopic regions.

I find the conclusions about the impact of immune selection on the viruses interesting, but I’m not completely sure how it is supported by the data- be interested to have more discussion of this.

Reviewer #2: Inconsistency between the legend (line 519) and Figure 2 B; DENV-1 III or I?

Reviewer #3: Page 6 – I believe it should say ‘Dataset C’ and not ‘Subset C’

**Part II – Major Issues: Key Experiments Required for Acceptance**

Reviewer #2: - Authors should clearly describe the sample size's representative-ness.

- Manuscript should include a more detail epidemiological data and explain whether there is direct correlation of specific genotype/lineages with the outbreaks.

- Whether the data presented can be used to predict future outbreak should be discussed.

- The differences of the vaccine strains with India's strains should be compared with other countries in terms of possible low efficacy caused by genotype differences.

- Limitations should be clearly described.

Reviewer #3: The largest limitation in the paper is that the authors make strong conclusions about the antigenic evolution of the serotypes, however, there is no direct measurement of antigenicity (such as neutralisation). Instead, they rely on the presence of nucleotide changes in previously identified epitopes, without being able to characterise what those changes actually do to neutralisation (if anything). I would suggest the authors discuss this limitation, and tone down the language when making conclusions about antigenic evolution.

The last sentence in the abstract concludes that the high incidence and pre-existing immunity are shaping dengue evolution. However, there was no attempt to reconstruct population immune profiles in the paper, so I’m not sure the authors can really conclude this.

A major conclusion from the paper is that the genetic distance (as measured by Hammings distance) between the E-gene of strains has fluctuated with 3-5 year periodicities. The authors conclude this is linked to population immunity changes. I think this is an interesting observation but I’m not sure the authors can conclude that these fluctuations are due to immunity. To better assess a potential role of immunity, the authors could repeat the analysis with the rest of the genome. If the changes in the E-gene were really driven by immunity, these fluctuations would be absent in parts of the genome that are not antigenically relevant.

On page 13 the authors say that DENV4-1 originated from the Philippines. I would argue this is very difficult to conclude as there are very few DENV4 sequences available from the region and it is very possible that the origin is in an unsampled location.

I found Figure 4C very difficult to read/interpret. I’m afraid I don’t have a better suggestion but anything that helps guide the reader interpret the plot would be useful.

On page 17, it was unclear to me what ‘dominant’ amino acids were, and what a >50% of the differences therefore actually means? What is the denominator here?

Again just because they are genetically different from vaccine strains, doesn’t necessarily mean there will be reduced efficacy.

In the final paragraph of the discussion, the authors conclude that their analysis shows that the immunity developed by the vaccine can also impact the future course of dengue evolution. While this may be true, I could not see how their analysis has shown this.
---

## [Decision Letter · Decision Letter 1]

17 Mar 2023

Dear Dr. Roy,

We are pleased to inform you that your manuscript 'Evolutionary dynamics of dengue virus in India' has been provisionally accepted for publication in PLOS Pathogens.

Best regards,

Leah Katzelnick

Guest Editor

PLOS Pathogens

Sonja Best

Section Editor

PLOS Pathogens

Kasturi Haldar

Editor-in-Chief

PLOS Pathogens

orcid.org/0000-0001-5065-158X

Michael Malim

Editor-in-Chief

PLOS Pathogens

orcid.org/0000-0002-7699-2064

Reviewer Comments (if any, and for reference):

Reviewer's Responses to Questions

**Part I - Summary**

Reviewer #2: The authors have answered/responded the comments raised by reviewer.

Reviewer #3: The authors have done a good job of responding to my concerns

**Part II – Major Issues: Key Experiments Required for Acceptance**

Reviewer #2: (No Response)

Reviewer #3: (No Response)

**Part III – Minor Issues: Editorial and Data Presentation Modifications**

Reviewer #2: The authors should use the standardized nomenclature of dengue virus (according to ICTV), i.e. DENV-1; DENV-2; DENV-3; DENV-4 (with a dash between DENV and number).

Reviewer #3: (No Response)

PLOS authors have the option to publish the peer review history of their article (what does this mean?). If published, this will include your full peer review and any attached files.

Reviewer #2: No

Reviewer #3: No

---

## [Editor Report · Acceptance letter]

29 Mar 2023

Dear Dr. Roy,

We are delighted to inform you that your manuscript, "Evolutionary dynamics of dengue virus in India," has been formally accepted for publication in PLOS Pathogens.

Best regards,

Kasturi Haldar

Editor-in-Chief

PLOS Pathogens

orcid.org/0000-0001-5065-158X

Michael Malim

Editor-in-Chief

PLOS Pathogens

orcid.org/0000-0002-7699-2064